# VidMuse: A Simple Video-to-Music Generation Framework with Long-Short-Term Modeling

## Abstract

In this work, we systematically study music generation conditioned solely on the video. First, we present a large-scale dataset by collecting 360K video-music pairs, including various genres such as movie trailers, advertisements, and documentaries. Furthermore, we propose VidMuse, a simple framework for generating music aligned with video inputs. VidMuse stands out by producing high-fidelity music that is both acoustically and semantically aligned with the video. By incorporating local and global visual cues, VidMuse enables the creation of coherent music tracks that consistently match the video content through Long-Short-Term modeling. Through extensive experiments, VidMuse outperforms existing models in terms of audio quality, diversity, and audio-visual alignment. Samples of results and comparisons with other methods are available at our anonymous project URL: `https://anonymous-project-demo.github.io/Anonymous_VidMuse/`.

## 1 Introduction

Music, as an essential element of video production, can enhance humans' feelings and convey the theme of the video content. Along with the development of social media platforms *i.e.*, YouTube and TikTok, some studies (Ma, 2022; Dasovich-Wilson et al., 2022; Millet et al., 2021) have shown that a piece of melodious music can vastly attract the audience's attention and interest in watching the video. It thus leads to a great demand for studying video-to-music generation (Di et al., 2021; Kang et al., 2023; Su et al., 2023; Gan et al., 2020; Hong et al., 2017; He et al., 2024).

Nevertheless, music creation for video is a challenging task, which need to understand both music theory and video semantics. It would be very time-consuming to produce a piece of suitable music for video in a hand-crafted manner. Therefore, it is desirable when we can automatically generate high-quality music for different genres of videos.

Currently, most of works (Copet et al., 2024; Huang et al., 2018; Yang et al., 2023b; Forsgren & Martiros; Huang et al., 2023; Schneider et al., 2023) have made significant achievements, especially in text-to-music generation, but the video-to-music generation still remains to be further studied. Specifically, existing works on video-conditioned music generation mainly focus on specific scenarios, such as dance videos (Li et al., 2021; Zhu et al., 2022), or on the symbolic music, *i.e.*, MIDI (Wang et al., 2020; Di et al., 2021; Zhuo et al., 2023; Kang et al., 2023). However, these works are unable to generate more diverse musical styles and are also difficult to generalize to various video genres. Moreover, Hong *et al.* (Hong et al., 2017) build a music–video retrieval dataset from YouTube-8M (Abu-El-Haija et al., 2016), albeit with limited video genres. Despite that there are also some prominent works (Hussain et al., 2023; Su et al., 2023) employing multi-modal inputs to generate music for the video, it is still worth studying *whether it is possible to generate high-quality and harmonious music for diverse genres of videos, conditioned solely on the visual input.*

Motivated by this, we first construct a large-scale dataset termed *V2M*, equipped with a comprehensive benchmark to evaluate the state-of-the-art works thoroughly. The video-music pairs are collected from YouTube with various genres, *e.g.*, movie trailers, advertisements, documentaries, vlogs, *etc*. In order to ensure the quality of our dataset, we establish a multi-step pipeline illustrated in Fig. 3 to systematically clean and pre-process data. The videos with low quality or composed of static images are filtered out. The proposed dataset contains three subsets: *V2M-360K* for pretraining, *V2M-20K* for finetuning, and *V2M-bench* for evaluation. We believe that *V2M* is able to facilitate the development of video-to-music generation.

Furthermore, on top of V2M, we propose a simple yet effective method, termed as **VidMuse**, to generate music only conditioned on the visual input. Instead of predicting the intermediate musical symbols such as MIDI or retrieving the music from the database, the proposed VidMuse, integrates both local and global visual clues to generate background music consistent with the video in an end-to-end manner. The core techniques in our method are a *Long-Short-Term Visual Module (LSTV-Module)* and a *Music Token Decoder*. Specifically, the LSTV-Module aims to learn the spatial-temporal representation of videos, which is the key to generating music aligned with the video. On the one hand, the long-term module models the entire video, capturing global context to understand the whole video. It contributes to the coherence of generated music at the video level. On the other hand, the short-term module focuses on learning the fine-grained clues at the clip level, which plays an important role in generating diverse music. The integration of two modules can improve the quality and visual consistency of generated music. In addition, the Transformer-based music token decoder is an autoregressive model, converting video embeddings obtained by LSTV-Module into music tokens. We formulate music generation as a task of next token prediction, which has been widely validated by the NLP community. The predicted music tokens are further decoded into the music signals by a high-fidelity neural audio compression model.

In summary, the main contributions of our work are as follows:

(a) We construct a large-scale video-to-music dataset, *i.e.*, *V2M*, which contains about 360k video-music pairs with high quality, covering various genres and themes. To the best of our knowledge, this is the largest and most diverse dataset for this task, which can facilitate future research.

(b) We propose a simple yet effective method, VidMuse, for video-to-music generation. The proposed method integrates both local and global cues in the video, enabling the generation of high-fidelity music tracks that are not only musically coherent but also semantically aligned with the video content.

(c) We benchmark several state-of-the-art works against our method on the V2M-bench via a series of subjective and objective metrics for a thorough evaluation. As demonstrated in experiments, VidMuse achieves state-of-the-art performance on *V2M-bench*, outperforming existing models in terms of audio quality, diversity, and audio-visual consistency.

## 2  RELATED WORK

In this section, we review the existing works related to video-to-music generation, which mainly fall into four categories:

**Video Representation.** Various methods have been proposed to learn the spatio-temporal representation (Liu et al., 2020b; 2021; Tran et al., 2018; Feichtenhofer et al., 2019; Arnab et al., 2021; Tong et al., 2022; Zhang et al., 2023; Liu et al., 2022) for videos. They aim to capture the contextual features of video frames, which is beneficial for video understanding. Recent advances primarily concentrate on developing video transformers, including ViViT (Arnab et al., 2021; Tong et al., 2022; Liu et al., 2022). These transformer-based methods achieve superior generalized performance on various video understanding tasks, such as video classification and temporal action localization. Among them, Tong *et al.* (Tong et al., 2022) extend masked autoencoders (He et al., 2022) from the image to the video, exhibiting the strong generalized performance in downstream tasks. Benefiting from the advance in multi-modal large language models, lots of works (Zhang et al., 2023; Lin et al., 2023; Munasinghe et al., 2023) of interactive video understanding have been proposed, which built upon the large language models (LLMs) (Touvron et al., 2023; Zheng et al., 2024; Raffel et al., 2020; Taori et al., 2023) and showcase the visual reasoning capabilities for video understanding.

**Audio-Visual Alignment.** Audio-visual alignment (Akbari et al., 2021; Rouditchenko et al., 2020; Shi et al., 2022; Cheng et al., 2022; Gong et al., 2022; Zhu et al., 2023; Yang et al., 2023c; Wang et al., 2023b; Wu et al., 2023a; Xing et al., 2024) aims to align the feature between audio, vision in the semantics level, which plays a vital role in tasks of audio-visual understanding and generation. For example, CAV-MAE (Gong et al., 2022) is an audio-visual MAE by integrates the contrastive learning and masked modeling method. Currently, many works go beyond exploring audio-visual alignment. ImageBind (Girdhar et al., 2023) extends CLIP (Radford et al., 2021) to more modalities, including audio, depth, thermal, and IMU data, which paves the way for cross-modal retrieval and generation. In addition, Wu *et al.* (Wu et al., 2023a) employ LLMs with multi-modal adaptors to support any modal data as input and output, showing strong capabilities in universal multi-modal understanding. These

methods transcend audio-visual alignment and dramatically advance the development of multi-modal representation learning.

**Conditional Music Generation.** Despite that there are lots of methods (Dong et al., 2018; Hawthorne et al., 2018; Huang et al., 2018; Liu et al., 2020a; Mittal et al., 2021; Lv et al., 2023; Maina, 2023) studying unconditional music generation, in this paper, we mainly focus on reviewing the methods of conditional music generation, which are more related to our work. Many researchers (Schneider et al., 2023; Agostinelli et al., 2023; Yang et al., 2023b; Forsgren & Martiros; Huang et al., 2023; Copet et al., 2024; Yuan et al., 2024; Deng et al., 2024) make their endeavours on text-to-music generation. Similar to Stable Diffusion (Rombach et al., 2022), these works (Yang et al., 2023b; Forsgren & Martiros; Huang et al., 2023; Schneider et al., 2023) try to adapt diffusion models for music generation. M$^2$UGen (Hussain et al., 2023) is a multi-modal music understanding and generation system that leverages large language models to process video, audio, and text. Video2Music(Kang et al., 2023) can generate music that matches the content and emotion of a given video. Moreover, the proposed V2Meow (Su et al., 2023) and MeLFusion (Chowdhury et al., 2024) conditioned on video and image, respectively, can generate music that supports style control via text prompts. In contrast to previous video-to-music works (Gan et al., 2020; Kang et al., 2023; Zhuo et al., 2023; Su et al., 2023), our VidMuse utilizes a short-term module and a long-term module to model local and global visual clues in videos. As a result, it can generate high-fidelity music aligned with the video.

**Video-to-music Datasets.** Many multi-modal datasets (Changpinyo et al., 2021; Morency et al., 2011; Tian et al., 2020; Lee et al., 2021; Gemmeke et al., 2017; Zhou et al., 2018; Miech et al., 2019; Srinivasan et al., 2021; Lin et al., 2014; Abu-El-Haija et al., 2016; Schuhmann et al., 2022; Hershey et al., 2017; Gemmeke et al., 2017) have been released, but there is still a lack of datasets for video-to-music generation. Hong *et al.* (Hong et al., 2017) construct the HIMV-200K with video-music pairs and aim to retrieve music for the video from the database. However, this dataset exhibits limited video genres and also suffers from the issue of data quality, as stated in (Zhuo et al., 2023). We observe that several works (Wang et al., 2020; Di et al., 2021; Zhuo et al., 2023; Li et al., 2024) aim to facilitate MIDI music generation. However, we argue this musical form imposes restrictions on diversity for the music generation. Other datasets (Zhu et al., 2022; Li et al., 2021) focus on generating music for dance videos only and have limited data size, which limits their applicability for general video-to-music models. As a result, this work constructs a large-scale video-to-music dataset where the music directly in wav format is diverse. We establish a rigorous pipeline for data collection and cleaning, which ensures the quality and diversity of our dataset. We expect the model can learn the music with more diverse forms from this dataset.

## 3 DATASET

In this section, we build a multi-step pipeline to clean and process source videos from YouTube to ensure data quality. After that, we construct a large-scale video-to-music generation dataset, *i.e.*, *V2M*, with a benchmark. The constructed dataset stands out for its significant size, high quality, and rich diversity, including a wide range of genres such as movie trailers, advertisements, documentaries, vlogs, *etc*. This comprehensive and diverse dataset aims to facilitate the video-to-music generation.

### 3.1 DATASET COLLECTION

To quickly collect a large scale of video-music pairs, we curate a series of query sets to retrieve corresponding videos from YouTube. In addition, we found that the music in the movie trailer usually showcases rich diversity and high quality. Therefore, we also aggregate a vast array of video information from the IMDb Non-Commercial Datasets (imd), including video types, names, release dates, etc. Queries are formulated based on the titles of these selections and retain the videos released after 2000, as videos from earlier periods are less likely to be of good quality. In the process of video crawling, we only keep the top 2 search results, resulting in a collection of around 400K videos, ranging from movies to documentaries. Besides, several existing datasets already contain video-music pairs, such as HIMV-200K (Hong et al., 2017), subsets of YouTube-8M (Abu-El-Haija et al., 2016) labeled with "Music" and "Trailer" tags. We incorporate these datasets into our collection to further expand its scope. After merging all sources, our final dataset comprises about 600K videos, spanning diverse genres and categories.

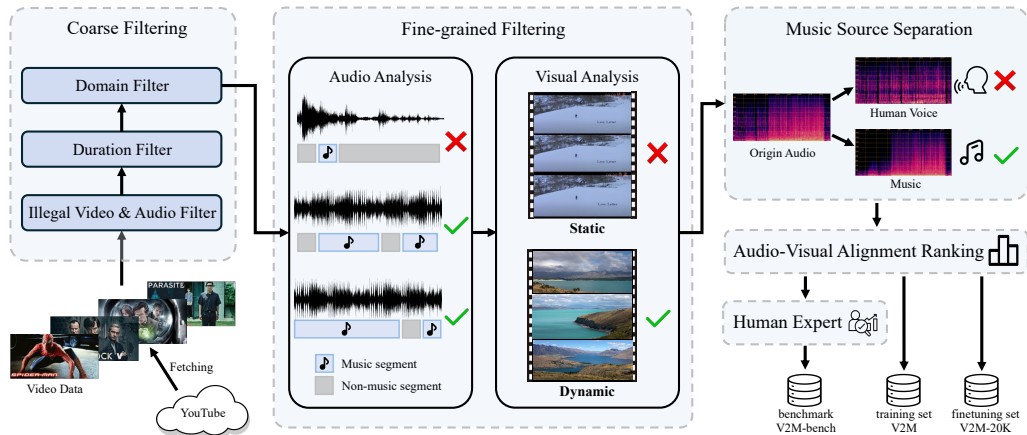

Figure 1: **Dataset Construction.** To ensure data quality, *V2M* goes through rule-based coarse filtering and content-based fine-grained filtering. Music source separation is applied to remove speech and singing signals in the audio. After processing, human experts curate the benchmark subset, while the remaining data is used as the pretraining dataset. The pretrain data is then refined using Audio-Visual Alignment Ranking to select the finetuning dataset.

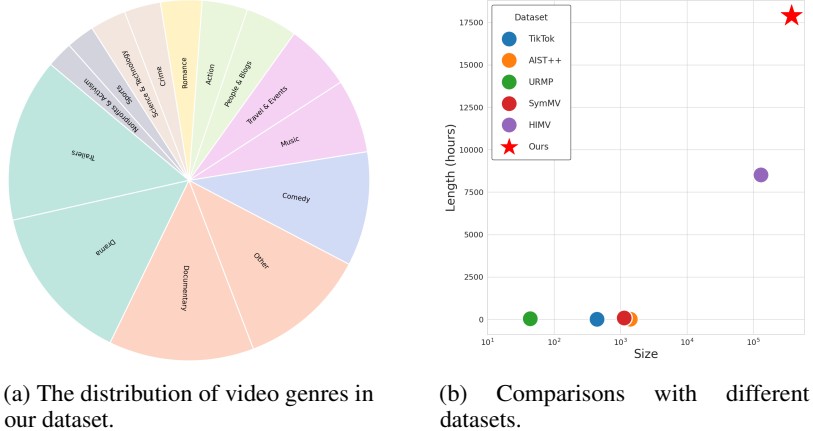

(a) The distribution of video genres in our dataset.

(b) Comparisons with different datasets.

Figure 2: **The statistics of our dataset.**

## 3.2 DATASET CONSTRUCTION

The 600K raw videos may include many low-quality samples. To address this, we develop a series of rigorous steps to filter out undesirable data and obtain a clean set. The overall pipeline of data processing is depicted in Fig. 1. The following steps outline our approach: (1) The process begins with coarse filtering, where we remove videos lacking audio or video tracks, videos that are too short or too long, those containing inappropriate content such as violence or explicit material, and those from categories like *Interview* and *News*, which generally have background music not aligned with the visual content. (2) Following this, we perform fine-grained filtering to retain videos with substantial music content and dynamic visual elements. We use an audio analysis model (Kong et al., 2020) to identify music segments, ensuring a sufficient portion of the audio is classified as music. In parallel, we analyze the visuals (Wang et al., 2004) to exclude videos consisting mainly of static images. (3) To further refine the dataset, we apply music source separation (Défossez et al., 2019) to isolate the music component by removing vocal tracks, enhancing the overall audio quality. (4) Finally, we rank the videos based on their audio-visual alignment scores (Girdhar et al., 2023) to ensure a high level of semantic correlation between the audio and visual modalities. The resulting videos are then split into training (*V2M*), fine-tuning (*V2M-20K*), and benchmark (*V2M-bench*) subsets. For details on dataset construction, please refer to the Appendix A.2).

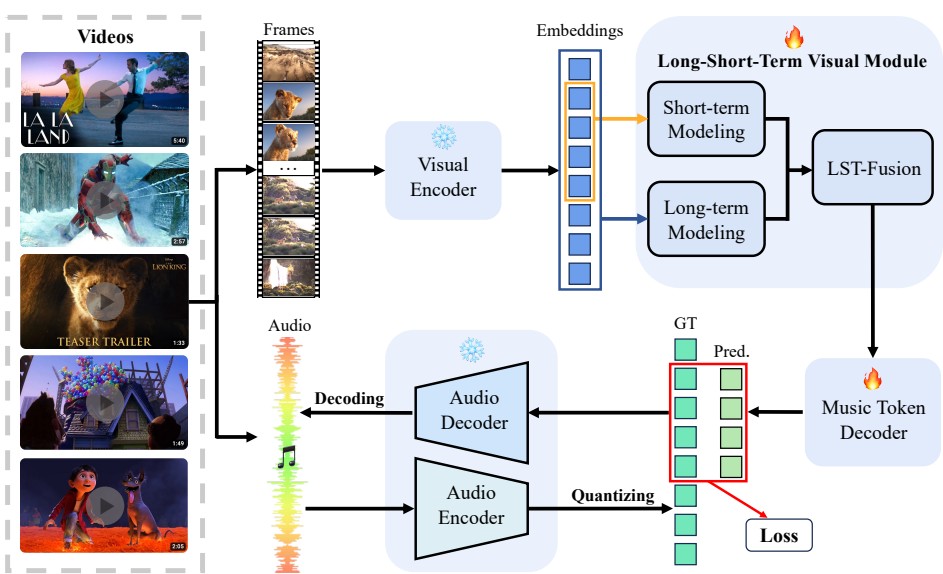

Figure 3: **Overview of the VidMuse Framework.**

## 3.3 DATA ANALYSIS

The above data pipeline yields three data splits. Specifically, the training set comprises ∼360K video-music pairs, totaling around $1.8 \times 10^4$ hours. The finetuning dataset consists of ∼20K pairs, totaling around $6 \times 10^3$ hours. The benchmark dataset contains 300 pairs, with a cumulative duration of 9 hours. Fig. 2a showcases the genre distribution of our training data, highlighting its comprehensive diversity. This diversity ensures a rich and varied dataset for the model training. As shown in Fig. 2b, we compare with other related datasets, demonstrating its advantage in data scale. Table 1 presents the number of samples and the total length for the training set, fine-tuning set, and test set.

**Dataset Necessity.** Some existing video-music pair datasets have been released (Hong et al., 2017; Wang et al., 2020; Di et al., 2021; Zhuo et al., 2023; Zhu et al., 2022; Li et al., 2021), but some of them (Wang et al., 2020; Di et al., 2021; Zhuo et al., 2023) aim to facilitate MIDI music generation, which limits the form of music. Datasets like (Zhu et al., 2022; Li et al., 2021) focus on generating music for dance videos only and have limited data size. The dataset constructed by (Hong et al., 2017) includes video-music pairs but exhibits limited video genres and suffers from data quality issues. Motivated by this, we develop the multi-step pipeline and curate a large-scale dataset *V2M* for the video-to-music generation.

Table 1: **The statistics of each subset.**

| Subset | #Samples | Length (Hours) |
|---|---|---|
| V2M | 360K | $1.8 \times 10^4$ |
| V2M-20K | 20K | $6 \times 10^3$ |
| V2M-bench | 300 | $9 \times 10^0$ |

## 4 METHOD

### 4.1 ARCHITECTURE OF VIDEMUSE

In this section, we elaborate on the proposed VidMuse, which leverages **LSTV-Module** to generate music that is aligned with video content. The framework's pipeline is shown in Fig. 3, which includes (1) a Visual Encoder, (2) a LSTV-Module, (3) a Music Token Decoder, and (4) an Audio Codec Decoder.

**Visual Encoder.** To generate music conditioned on the video, we first need to extract the high-level features from a stack of frames. Given an input video, the visual encoder extracts feature representations $\mathbf{X} \in \mathbb{R}^{N \times P \times D}$. Here, N is the number of input frames, $P$ refers to the sequence

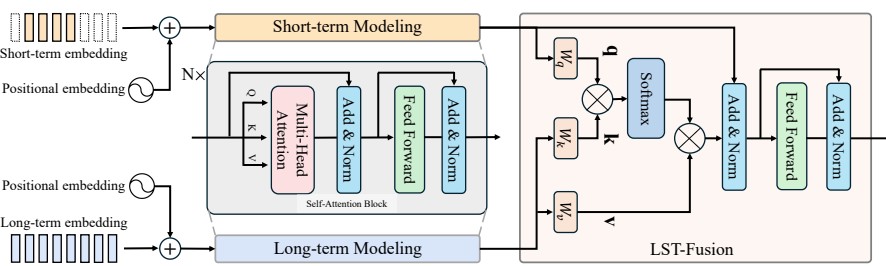

Figure 4: **Long-Short-Term Visual Module.**

length with the class token, and $D$ denotes the size of the feature vectors. Currently, there are lots of visual encoders available, including 2D (Dosovitskiy et al., 2020), 3D (Tong et al., 2022; Arnab et al., 2021) and multi-modal (Radford et al., 2021) models, which will be validated in the Sec.5.5.

**LSTV-Module.** Generating music for videos with variable length still presents significant challenges, especially for a long video. Because sometimes it is difficult to directly model on whole video due to hardware limitations. Prior studies opt to generate music in segment level (Kang et al., 2023; Di et al., 2021; Hussain et al., 2023). However, such a manner often lacks sufficient context information. Music, when generated based on visual content, varies in expression depending on the context. Even the same video segment may lead to distinct musical interpretations when being in different contexts. By incorporating global guidance, it can enhance the alignment of the generated music with the overall video content. As a result, to capture both local and global visual clues, the visual features extracted from the visual encoder are fed into the LSTV-Module. As depicted in Fig. 3, the Short-term Module takes segment-level embeddings as input, aiming to capture local dependencies $\mathbf{X}_s \in \mathbb{R}^{N_s \times P \times D}$ to ensure that the generated music aligns with short-term variations in the video, while Long-term Module models on video-level embeddings, providing context $\mathbf{X}_l \in \mathbb{R}^{N_l \times P \times D}$ to guide the short-term module in generating more visually coherent music. $N_s$ and $N_l$ is the number of frames sampled from the video.

The architecture of the LSTV-Module is shown in Fig. 4. To capture both global and local visual clues, we leverage the self-attention mechanism (Vaswani et al., 2017) in both Long-term and Short-term Modeling. Long-term modeling extracts long-range dependencies, while short-term modeling focuses on local details. This results in refined long-term features $\mathbf{F}_l \in \mathbb{R}^{N_l \times P \times D}$ and short-term features $\mathbf{F}_s \in \mathbb{R}^{N_s \times P \times D}$.

To incorporate global guidance for generating segment-based music, we design LST-Fusion. It integrates long-term and short-term features by utilizing the cross-attention mechanism $\mathrm{CA}(\cdot)$ with Query ($\mathbf{Q}$), Key ($\mathbf{K}$), and Value ($\mathbf{V}$), which can be mathematically formulated as:

$$\mathbf{Z}' = \mathrm{CA}(\mathbf{Q}, \mathbf{K}, \mathbf{V}), \text{ where } \mathbf{Q} = \mathbf{F}_s, \mathbf{K} = \mathbf{F}_l, \mathbf{V} = \mathbf{F}_l, \tag{1}$$

This mechanism allows the model to query global information rather than generating music based solely on local visual features. It guarantees that the generated music is more consistent with the overall video content. After the cross-attention, a linear layer projects $\mathbf{Z}'$ to $\mathbf{Z} \in \mathbb{R}^{N_s \times P \times M}$, where $M$ represents the input vector dimension of the music token decoder in the next step.

**Music Token Decoder.** We adopt an autoregressive approach to predict the music tokens $\bar{\mathbf{Y}}$ conditioned on the video segment. Music token decoder is implemented by a transformer decoder with a linear classifier. We set the latent vector size of the transformer decoder to $M$, allowing to scale up or down the model's size. The decoder incorporates a cross-attention mechanism that receives the visual signal $\mathbf{Z} \in \mathbb{R}^{N_s \times P \times M}$, where $N_s$ is the number of frames sampled in the video segment. At each time step $t$ (where $t = 1, \ldots, T$), the decoder predicts the logits of current token $\bar{\mathbf{Y}}_t \in \mathbb{R}^{K \times C}$ based on previous tokens and visual context $\mathbf{Z}$. Here, $K$ denotes the number of codebooks, and $C$ represents the vocabulary size.

**Audio Codec.** The audio codec can convert an audio segment into discretized codebooks and, conversely, decode codebooks back into audio. The size of codebooks is $K \times T$, where $T$ denotes the length of the video. In the training stage, we need to encode the ground truth audio $\mathbf{A}$ into discretized

codebooks that serve as supervise signals for the next token prediction. After training finished, given a video, we use the autoregressive fashion to predict codebooks for all time steps and decode the predictions into a piece of music.

## 4.2 DISCUSSION

In this section, we introduce VidMuse, a simple yet effective framework for video-to-music generation, which offers several key advantages: (1) Our framework is highly adaptable, with each module designed for smooth integration, facilitating the identification of optimal settings. Additionally, VidMuse can infer long videos via a sliding window approach (see Appendix A.4). 2) In contrast to previous approaches (Kang et al., 2023; Di et al., 2021; Hussain et al., 2023) that primarily focus on short-term visual cues, our method leverages both short-term and long-term visual information. It allows for the understanding of local semantics, producing music that is consistent with the video's context. Furthermore, instead of predicting intermediate text embeddings or musical symbols such as MIDI, our approach directly generates background music that is coherent with the video in an end-to-end manner. 3) As demonstrated Table 2, the KL metric indicates that our method produces video-level music generation results that are more aligned to the ground truth. Furthermore, the ImageBind metric shows that our generated music is more semantically aligned with the visual content overall. These results validate the overall effectiveness of our framework.

## 5 EXPERIMENTS

In this section, we first elaborate on the implementation details of our experiment. Next, we introduce the evaluation metrics used to assess the performance and then present the main results for both our method and the baseline methods using objective metrics and a subjective user study. Then, we demonstrate that our VidMuse exhibits impressive performance on all metrics. Furthermore, we conduct a comprehensive ablation study to explain our design choices and their impact on the performance.

### 5.1 IMPLEMENTATION DETAILS

We choose CLIP (Radford et al., 2021) as the visual encoder in default (see experiments in Sec. 5.5). Since this work does not focus on audio encoding and decoding, we use Encodec (Défossez et al., 2019) for 32 kHz monophonic audio as our default compression model and use the pretrained transformer model proposed in MusicGen (Copet et al., 2024). The training stage utilizes the AdamW optimizer (Loshchilov & Hutter, 2017) for 56,000 steps with a batch size of 240 samples. The hyperparameters are set to $\beta_1 = 0.9$, $\beta_2 = 0.95$, with a weight decay of 0.1 and gradient clipping at 1.0. A cosine learning rate schedule is employed, incorporating a warm-up phase of 4,000 steps and an exponential moving average decay of 0.99. The training is completed on H800 GPUs, with float16 mixed precision. This setup leverages 360K samples for pre-training and 20K samples for fine-tuning. A top-k strategy is applied for sampling, retaining the top 250 tokens with a temperature setting of 1.0. In the inference stage, we set the sliding window size as 30s, and the length of the window's overlap as 0.5s.

### 5.2 EVALUATION METRICS

To quantitatively evaluate the effectiveness of our model, we employ a series of metrics to assess different models in terms of quality, fidelity, and diversity of the generated music. These metrics include the Frechet Audio Distance (FAD)[1] (Kilgour et al., 2018), Frechet Distance (FD)[1], Kullback-Leibler Divergence (KL)[1], as well as Density and Coverage[2] (Naeem et al., 2020). Additionally, we utilize the ImageBind Score[3] (Girdhar et al., 2023) to examine the alignment between the video and the generated music. We acknowledge that ImageBind has limitations as it is not specifically trained on music data, but it currently seems to be a possible option for evaluating the semantic

---

[1] https://github.com/haoheliu/audioldm_eval

[2] https://github.com/clovaai/generative-evaluation-prdc

[3] https://github.com/facebookresearch/ImageBind

Table 2: **Comparison with naive baselines and state-of-the-art methods.**

| Methods | Metrics | | | | | |
|---|---|---|---|---|---|---|
| | KL ↓ | FD ↓ | FAD ↓ | density ↑ | coverage ↑ | Imagebind ↑ |
| GT | 0.000 | 0.000 | 0.000 | 1.167 | 1.000 | 0.241 |
| Caption2Music | 1.081 | 40.199 | 2.485 | 0.378 | 0.486 | 0.191 |
| Video2Music (Kang et al., 2023) | 1.782 | 144.881 | 18.722 | 0.103 | 0.023 | 0.136 |
| CMT (Di et al., 2021) | 1.220 | 85.704 | 8.637 | 0.080 | 0.070 | 0.124 |
| M$^2$UGen (Hussain et al., 2023) | 0.997 | 52.246 | 5.104 | 0.608 | 0.433 | 0.181 |
| VM-NET* (Hong et al., 2017) | 0.899 | 67.480 | 6.252 | 0.986 | 0.383 | 0.147 |
| VidMuse | **0.734** | **29.946** | **2.459** | **1.250** | **0.730** | **0.202** |

alignment between video and generated music. More details about these metrics are provided in the Appendix A.3.

## 5.3 MAIN RESULTS

We benchmark several state-of-the-art methods, serving as baselines to compare with our method: 1) **Caption2Music** serving as a naive baseline, which employs the method (Spa, 2022) to extract the video captions and then outputs the music by feeding captions into MusicGen (Copet et al., 2024). 2) **Video2Music** (Kang et al., 2023) and 3) **CMT** (Di et al., 2021) which botch focus on predicting MIDI notes (Rothstein, 1995) from videos while our method directly generates music signals. 4) **M$^2$UGen** (Hussain et al., 2023), a strong baseline, which leverages a language model to connect vision and language, then use MusicGen (Copet et al., 2024) to generate music from language. 5) **VM-NET** (Hong et al., 2017), essentially different from the above methods, retrieves the piece of music from the database for a given video, while other methods predict music by learning from video-music pairs.

In Table 2, VidMuse, with both global and local visual modeling, exhibits impressive performance on all metrics. Specifically, compared with Video2Music (Kang et al., 2023) and CMT (Di et al., 2021), VidMuse shows the superiority in the diversity of generated music based on the density or coverage, which justifies the advantage of directly predicting music signals compared with MIDI notes. In addition, our method even outperforms the strong competitors, *i.e.*, M$^2$UGen. It proves that predicting music directly based on video input can also achieve better performance. Furthermore, compared with a retrieval-based method *i.e.*, VM-NET (Hong et al., 2017), VidMuse achieves a higher Imagebind score, indicating that the music generated by the learning-based method is more consistent with the video semantics.

## 5.4 USER STUDY

In our user study, we randomly sample 600 video-music pairs from the benchmark to conduct an A/B test, which is a widely used subjective evaluation method in the music field (Donahue et al., 2023; Yuan et al., 2024), across five music generation methods: CMT, M$^2$UGen, Caption2Music, Ground Truth (GT), and VidMuse. The test was distributed among 40 participants, ensuring each method was compared against another 60 times. The evaluation criteria are four-fold: 1) **Audio quality**: Refers to the sound quality of the audio track; 2) **Video-music alignment**: Assesses how well the music matches the visual content, e.g., a scene showing a woman crying should ideally be paired with music that sounds sad; 3) **Musicality**: Evaluates the aesthetic quality of the music, distinct from audio quality. For example, a piece of music may have good audio quality, but if it is out of tune, it would be considered to have poor musicality; 4) **Overall assessment**: Comprehensively evaluates the performance for models.

The user study process is shown in Fig. A1. Participants are asked to choose the better sample for each criterion. Fig. 5 illustrates the statistical results of the user study, where the value at matrix[$i$][$j$] ranges from 0 to 100, indicating the % of times listeners preferred the method in $i$-row compared to the method in $j$-column. For example, in Fig. 5 (c), the value of matrix[2][4] represents that VidMuse outperforms CMT in 77% of cases in terms of Musicality. Across all criteria, our method surpasses

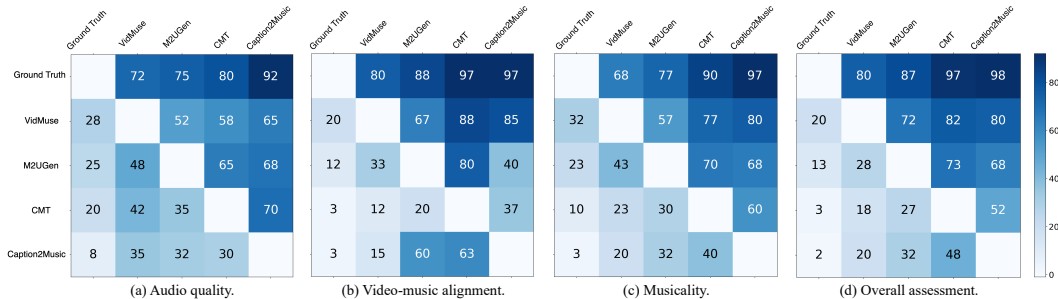

Figure 5: A/B test user study results. Please refer to Sec. 5.4, where we introduce four criteria, *i.e.*, audio quality, video-music alignment, musicality, and overall assessment.

others in more than half of the comparisons, except when compared to the ground truth. Overall, these results robustly validate VidMuse's effectiveness through subjective evaluation.

## 5.5 ABLATION STUDIES

In this section, we conduct comprehensive ablation studies to determine the optimal hyperparameters and design choices. First, we examine the impact of different model design choices on the generation quality. After that, we explore the impact of the visual encoders and model size on performance. More ablation results are in the Appendix A.1.

**Justification of Design Choices.** To validate the impact of different model design choices on our generation results and verify the effectiveness of our method, we first design two modules: a short-term modeling module (STM) and a long-term modeling module (LTM). VidMuse-STM aims to ablate the contribution of STM by using only STM without LTM, while VidMuse-LTM utilizes only LTM. We present the results in Table 3. Based on these results, we gain the insight that local information plays a more important role in the generation. By integrating global guidance with local information, we improve the alignment of the generated music with the overall video content. Second, we implement two variants with **C**ross-**A**ttention with learnable **Q**ueries (**CAQ**) in our framework. Specifically, CAQ_SL first uses a **CAQ** where $K$ and $V$ are short-term features and then uses a **CAQ** where $K$ and $V$ are long-term features. CAQ_LS do it in an opposite order. As shown in Table 3, VidMuse outperforms two variants. It seems that the learnable query requires a deeper architecture, as demonstrated by DETR (Carion et al., 2020) and Q-Former (Li et al., 2023), but this manner reduces the model's efficiency. Furthermore, we test a baseline using Slowfast (Feichtenhofer et al., 2019) and find its performance degenerates in Table 3. This may be because the fast path uses a higher frame rate in inputs than the slow path, but their temporal receptive fields are identical. Thus, Slowfast still lacks global guidance.

**The impact of model size on performance.** For the architecture of the music token decoder, we test three variants with different scales: small (455M), medium (1.9B), and large (3.3B). The medium-sized VidMuse-M is recognized as the default setting if not stated, as it achieves the best results across all metrics, outlined in Table 4. We observe the larger VidMuse-L model fails to deliver proportional improvements. This discrepancy can be partly attributed to limited GPU resources. The model's performance is near saturation, and the larger model does not exhibit significant advantages. As a result, VidMuse-M is recognized as the optimal choice, achieving a trade-off between performance and efficiency.

**Visual Encoder.** We conduct ablation experiments to study the impact of various visual encoders. As shown in Table 5, we experiment with different visual encoders, including ViT (Dosovitskiy et al., 2020), CLIP (Radford et al., 2021), VideoMAE (Tong et al., 2022), and ViViT (Arnab et al., 2021). Our results show that VidMuse remains robust in processing visual information for music generation across all encoder choices. To balance computational efficiency and generation quality, we select CLIP (Radford et al., 2021) as the default visual if not stated.

Table 3: **Ablation studies on design choices.**

| Methods | Metrics | | | | | |
|---|---|---|---|---|---|---|
| | KL ↓ | FD ↓ | FAD ↓ | density ↑ | coverage ↑ | Imagebind ↑ |
| VidMuse-STM | 0.898 | 45.752 | 4.915 | 1.124 | 0.470 | 0.196 |
| VidMuse-LTM | 0.858 | 53.907 | 16.074 | 1.439 | 0.406 | 0.205 |
| VidMuse-CAQ_SL | 0.843 | 48.940 | 3.733 | 0.947 | 0.547 | 0.163 |
| VidMuse-CAQ_LS | 0.919 | 45.335 | 2.917 | 0.562 | 0.720 | 0.181 |
| VidMuse-Slowfast | 1.511 | 84.683 | 10.029 | 0.266 | 0.285 | 0.118 |
| VidMuse | **0.738** | **36.171** | **2.369** | **1.175** | **0.710** | **0.207** |

Table 4: **Ablation studies on model size.**

| Models | Params | Metrics | | |
|---|---|---|---|---|
| | | KL ↓ | FAD ↓ | density ↑ |
| VidMuse-S | 455M | 0.854 | 4.695 | 1.349 |
| VidMuse-M | 1.9B | **0.843** | **2.413** | **1.487** |
| VidMuse-L | 3.3B | 0.873 | 4.041 | 1.330 |

Table 5: **Ablation studies on video encoders.**

| Encoders | Metrics | | | |
|---|---|---|---|---|
| | KL ↓ | FD ↓ | density ↑ | GFLOPs ↓ |
| ViViT | 0.822 | **37.167** | **1.433** | 451.83 |
| VideoMAE | 0.778 | 37.900 | 1.074 | 360.99 |
| CLIP | **0.753** | 38.261 | 1.122 | **141.24** |
| ViT | 0.876 | 50.427 | 1.081 | 562.64 |

## 5.6 QUALITATIVE ANALYSIS

In Fig. A2, our qualitative analysis underscores specific limitations in the methods employed by CMT and $M^2$UGen. For CMT, the method's reliance on extracting only low-level visual cues for music generation, particularly in creating symbolic music, leads to some discontinuities. As illustrated in Fig. A2, "Abrupt gaps" occur, especially in darker video segments where the model fails to predict symbolic music notes, resulting in silence.

$M^2$UGen utilizes LLMs to fuse multimodal representation and then project LLMs' embeddings into music via a text-to-music generation model. However, the music generated by this work usually showcases repetitive musical themes and suffers from a lack of diversity. Additionally, $M^2$UGen is constrained to generating music segments of only 30 seconds, which necessitates splitting longer videos into clips for separate processing. Consequently, this leads to abrupt transitions between segments after stitching, compromising the overall musical continuity.

The last row of Fig. A2 showcases our Long-Short-Term (LST) approach, highlighting its ability to produce globally diverse music that captures the essence of the ground truth. This method ensures our music generation remains contextually rich and aligned with the video.

## 6 CONCLUSION

In this work, we build a rigorous pipeline to collect high-quality and diverse video-music pairs, curating a comprehensive dataset **V2M**. Then, we propose VidMuse, a simple yet effective method for video-to-music generation. Our approach utilizes a Long-Short-Term approach to capture both local and global visual clues in the video, allowing for the generation of contextually rich and musically diverse outputs. To validate our method, we benchmark a series of state-of-the-art methods as baselines to compare with VidMuse. Through comprehensive quantitative studies and qualitative analyses, our method has demonstrated its superiority over the existing methods.

**Limitations**. Our work achieves a significant advancement in video-to-music generation, but it still has some limitations. First, the current implementation relies on the EnCodec model (Défossez et al., 2022) trained on a dataset that restricts the system's capacity to a 32kHz audio sampling rate. This codec exhibits a noticeable reconstruction loss in the audio signal, potentially lowering the quality of the generated music. Second, training large models requires substantial computational resources. Our future work aims to overcome these limitations by integrating advanced codec technologies to enhance audio reconstruction fidelity, and exploring a more efficient method design.

## 7 ETHICS STATEMENT

Our research involves rigorous pipeline for audio-video data processing. As such, we take several steps to to avoid potential ethical concerns and inappropriate data usage. (1) To avoid non-compliant videos such as those containing explicit or violent content, our queries are from Imdb (imd), a platform with curated and reviewed content. The videos were collected from YouTube, which enforces community guidelines and employs automated detection systems to filter inappropriate material. Additionally, during the data construction, we excluded unsuitable content to ensure our dataset meets ethical requirements. (2) To comply with data copyright requirements, we do not publicly distribute video data but provide relevant metadata, such as YouTube IDs to the community. Our open-sourced models in the future will be trained exclusively on our private data. We commit to using all training data solely for research purposes and will not apply it commercially. In addition, our research outputs will be released under a Creative Commons license. Generally, with these measures, we aim to uphold ethical standards and conduct our research responsibly.

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

## A APPENDIX

### A.1 ADDITIONAL EXPERIMENTS

Additional experiments focusing on pretraining effects, visual encoders, and codebook patterns are provided in the appendix. These parts provide insight into the decision-making process for selecting the experimental configurations within the VidMuse framework.

**Pretraining Effect.** Our ablation study on the effects of the data scale during finetuning, as detailed in Table A1, highlights a balance between data size and model performance. Despite not performing best in all the metrics, the model finetuned with 20k pair data emerges as our choice. The 20k data offers a compelling trade-off: it significantly improves performance across key metrics without requiring the extensive computational resources that larger datasets demand.

**Exploration on model inputs.** To explore the impact of different video sampling rates and the duration of video segments in the Short-Term module on performance, we conducted ablation studies on input FPS and short-term segment duration, detailed in Table A2. To intuitively assess the effectiveness of different settings, we employ an **Average Rank** (AR) metric. The AR metric ranks the results for a metric across all methods within the same table. The ranking result is from 1 to $N$ (equals to the number of methods within the table), where 1 is the best and $N$ is the worst. We eventually obtain AR results by averaging the ranking results for all metrics. Note that the AR results cannot be compared across different tables since this metric is designed to showcase the dominance for each method within one table clearly. From Table A2, we observe that increasing both FPS and duration tends to enhance model capabilities, suggesting that denser frame sampling yields a more detailed video representation, thereby improving music generation. Nevertheless, to balance computational costs and performance, we use a 30-second duration at 2 FPS as our optimal setting.

**Codebook Pattern.** The exploration of codebook interleaving patterns has attracted attention from researchers across several domains (Zeghidour et al., 2021; Wang et al., 2023a; Copet et al., 2024; Yang et al., 2023a; Lan et al., 2023). In our ablation study focusing on the patterns, we find that while the Parallel and Vall-E (Wang et al., 2023a) patterns align with the findings for text-to-music generation in MusicGen (Copet et al., 2024), the flattened codebook pattern does not consistently exceed the performance of the delay pattern in tasks of generating music from video. The delay pattern, notable for its relatively low computational cost, is therefore selected for our implementation. The results of this study are presented in table A3.

### A.2 DETAILS OF DATASET CONSTRUCTION

**Coarse Filtering.** We design a rule-based filtering strategy for initial data screening. First, we perform illegal video and audio filters, which filter out the video without an audio track or a video track. Next, we apply a duration filter to filter out videos based on their duration, excluding those that are either too long (over 480 seconds) or too short (under 30 seconds). Additionally, we implement a domain filter to examine metadata and exclude specific categories such as *Interview*, *News*, and *Gaming*, which often have background music that lacks semantic alignment with the visual content. We also filter out videos containing inappropriate content, such as violence or explicit material.

**Fine-grained Filtering.** To further ensure the quality of our data, we conducted additional audio and visual analyses. For the audio analysis, raw videos may contain audio segments without music,

Table A1: **Ablation studies on the ratio of finetuning data.**

| Finetuning Data | Metrics | | | | | |
|---|---|---|---|---|---|---|
| | KL ↓ | FD ↓ | FAD ↓ | density ↑ | coverage ↑ | Imagebind ↑ |
| 0 | **0.712** | 38.184 | 3.956 | 1.125 | 0.583 | 0.181 |
| 10k | 0.717 | 34.667 | 2.961 | 0.856 | 0.673 | 0.196 |
| 20k | 0.734 | **29.946** | **2.459** | **1.250** | **0.730** | **0.202** |
| 40k | 0.776 | 41.075 | 3.557 | 1.094 | 0.726 | 0.195 |
| 60k | 0.828 | 40.160 | 2.844 | 0.977 | 0.660 | 0.192 |

Table A2: **Ablation studies on video duration and FPS.**

| Duration(s) | FPS | Metrics | | | | | | |
|---|---|---|---|---|---|---|---|---|
| | | KL ↓ | FD ↓ | FAD ↓ | density ↑ | coverage ↑ | Imagebind ↑ | AR ↓ |
| 5 | 2 | 0.820 | 51.101 | 4.117 | 1.430 | 0.74 | 0.148 | 7.00 |
| 15 | 2 | 0.849 | 41.131 | 2.709 | 1.406 | 0.803 | 0.181 | 5.33 |
| 30 | 2 | 0.843 | 41.354 | 2.413 | 1.487 | 0.840 | **0.193** | 3.67 |
| 5 | 4 | **0.800** | 51.540 | 4.343 | 1.271 | 0.787 | 0.145 | 7.17 |
| 15 | 4 | 0.830 | 41.154 | 2.562 | 1.278 | 0.823 | 0.176 | 5.17 |
| 30 | 4 | 0.849 | 40.032 | 2.418 | 1.538 | **0.843** | **0.193** | 2.84 |
| 5 | 8 | 0.819 | 50.667 | 4.069 | 1.515 | 0.743 | 0.153 | 5.67 |
| 15 | 8 | 0.857 | 42.106 | 2.790 | 1.476 | 0.753 | 0.187 | 6.00 |
| 30 | 8 | 0.824 | **38.942** | **2.299** | **1.573** | **0.843** | 0.180 | **2.17** |

such as speech, silence, *etc*. To ensure the final dataset consists of high-quality video-music pairs, we retain only those videos with a larger portion of music content. We utilize the sound event detection model PANNs (Kong et al., 2020), which provides frame-level event labels across the entire video to identify music events. Based on the observation from a subset of videos, we define two thresholds, *i.e.*, a confidence threshold and a duration threshold for analyzing the music event. The confidence threshold is set at $0.5$, indicating an audio frame is considered a music event if the PANNs model predicts the probability of the "Music" label to be over $0.5$. The duration threshold of a music event requires that at least $50\%$ of the audio's frames are classified as music events for the video to be considered valid.

For the visual analysis, some videos only consisting of static images will be removed. Specifically, we uniformly sample multiple temporal windows without overlap from the video. Within each window, we use Structural Similarity Index Measure (SSIM) (Wang et al., 2004) between the first frame and the last frame. By aggregating average SSIM values from all temporal windows, we remove the videos with average SSIM values lower than a threshold of 0.8, empirically.

**Music Source Separation.** Since the irrelevant human speech in videos poses a negative impact on music generation, we apply music source separation to process the videos. We employ Demucs (Défossez et al., 2019) as the music source separation model to filter out the speech signals.

**Audio-Video Alignment Ranking.** ImageBind-AV (Girdhar et al., 2023) scores usually reflect the semantic correlation between the vision and audio modality. To construct a high-quality subset with better alignment, we compute the ImageBind-AV scores for all the data and rank them accordingly.

After filtering and ranking, we split the final videos into the training set, *V2M*, from all the paired data. The top 20K pairs are selected to form the finetuning subset, *V2M-20K*. In addition, we randomly sample 1,000 videos excluded from the training set. These 1,000 videos are then further evaluated by five human experts based on audio quality and the degree of audio-visual alignment. Ultimately, the top 300 high-quality videos are selected as a test set, termed as *V2M-bench*.

Table A3: **Ablation studies on codebook pattern.**

| Patterns | Metrics | | | | | |
|---|---|---|---|---|---|---|
| | KL ↓ | FD ↓ | FAD ↓ | density ↑ | coverage ↑ | Imagebind ↑ |
| Parallel | 0.921 | 68.603 | 18.243 | 0.562 | 0.183 | 0.166 |
| Flatten | **0.819** | 52.931 | 4.260 | 1.351 | 0.500 | **0.201** |
| Delay | 0.843 | **41.354** | **2.413** | **1.487** | **0.840** | 0.193 |
| Vall-E | 0.866 | 57.286 | 4.681 | 1.148 | 0.354 | 0.189 |

## A.3 Details of Evaluation Metrics

**Frechet Audio Distance (FAD)** (Kilgour et al., 2018) is a reference-free evaluation metric for assessing audio quality. Similar to Frechet Image Distance (FID)(Heusel et al., 2017), it compares the embedding statistics of the generated audio clip with ground truth audio clips. A shorter distance usually denotes better human-perceived acoustic-level audio quality. However, this metric cannot reflect semantic-level information in audio. We report the FAD based on the VGGish(Hershey et al., 2017) feature extractor.

**Frechet Distance (FD)** measures the similarity between generated samples and target samples in audio generation fields. It's similar to FAD but uses a PANNs feature extractor instead. PANNs(Kong et al., 2020) have been pre-trained on AudioSet(Gemmeke et al., 2017), one of the largest audio understanding datasets, thus resulting in a more robust metric than FAD.

**Kullback-Leibler Divergence (KL)** reflects the acoustic similarity between the generated and reference samples to a certain extent. It is computed over PANNs' multi-label class predictions.

**Density and Coverage** (Naeem et al., 2020) measures the fidelity and diversity aspects of the generated samples. Fidelity measures how closely the generated samples match the real ones, while diversity assesses whether the generated samples capture the full range of variation found in real samples. We use CLAP(Wu et al., 2023b) embeddings for manifold estimation.

**Imagebind Score** (Girdhar et al., 2023) assesses to what extent the generated music aligns with the videos. Despite the fact that Imagebind extends the CLIP to six modalities, we only use the branches of audio and vision. Since we use ImageBind to filter out video-audio pairs with a low matching score during dataset construction, the ImageBind score is naturally used in our evaluation. We acknowledge that ImageBind is not specifically trained on music data, which may limit its effectiveness in capturing the full complexity of video-music alignment. However, it remains the most suitable option available for this task at present.

## A.4 Details of the Inference Process

When predicting music on videos of arbitrary length, maintaining music consistency and coherence is particularly important. However, it leads to a significant challenge on computational resources due to the quadratic dependency of transformers-based models on sequence length (Zaheer et al., 2020; Beltagy et al., 2020). To address this problem, we adopt a sliding window approach for inferring the whole video.

During inference, given an input video with a length of $L$, we define $L_s$ as the length of the sliding window and $O$ as the overlap between consecutive windows. With the window start position $t$ initially set to 0, the inference involves the following steps compactly while $t + L_s \leq L$: (1) using a visual encoder to extract feature representations $\mathbf{X}$ and capture long-term dependencies $\mathbf{X}_l$; (2) collecting embeddings within the window $[t, t + L_s]$ to obtain $\mathbf{X}_s$; (3) predicting the music tokens $\bar{\mathbf{Y}}$ for the reduced window $[t, t + L_s - O]$ based on $\mathbf{X}_l$ and $\mathbf{X}_s$; (4) decoding $\bar{\mathbf{Y}}$ to the predicted audio $\bar{\mathbf{A}}$ using the audio decoder; (5) move the window forward by setting $t = t + L_s - O$, and repeating steps (2) to (5) until the end of the video.

After finishing above steps, we can concatenate all musical segments to form a cohesive piece of music that aligns in duration with the video.

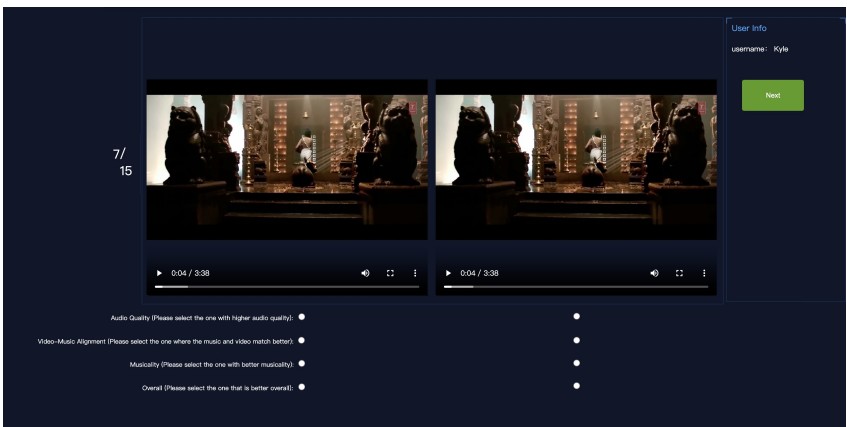

Figure A1: User study process. Participants evaluate the videos based on four criteria: Audio Quality, Video-Music Alignment, Musicality, and Overall Assessment.

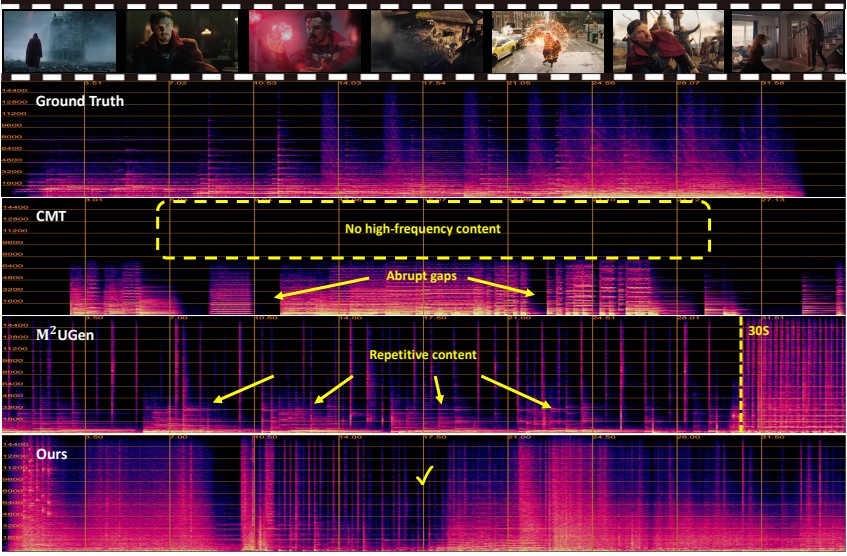

Figure A2: Qualitative Comparison results on sound spectrograms produced by different methods.

