# OpenReview forum: "VidMuse: A Simple Video-to-Music Generation Framework with Long-Short-Term Modeling"
_ICLR.cc/2025/Conference — ICLR 2025 Conference Withdrawn Submission_

### Official Review · Reviewer_DbJ5 · 2024-10-22

**Soundness:** 3
**Presentation:** 1
**Contribution:** 2
**Rating:** 5
**Confidence:** 5

**Summary:**

This paper presents a large video-music dataset with 360K samples and its curation process in detail. A new video-to-music generation method VidMuse is further proposed, which uses a long-short-term approach for local and global modeling to generate rich and diverse music tracks. The authors conduct objective and subjective experiments to validate the performance of the method.

**Strengths:**

1. The scale of the proposed dataset is significantly larger than previous video-music datasets. The data filtering process is also carefully designed and presented in detail.
2. The authors conduct sufficient ablation studies on each component of VidMuse. The generation quality is generally satisfying, judging from the demos.

**Weaknesses:**

1. The authors claim that their method is end-to-end while the MIDI-based methods are not end-to-end, but VidMuse relies on a pretrained audio decoder MusicGen to generate music from embeddings. Besides, M2UGen also uses a similar decoding strategy. The difference between VidMuse and M2UGen is explained as "conditioned solely on visual input", but M2UGen can generate music with only video inputs. It is doubtful that the performance gain mainly comes from the larger amount of training data. The comparison with Diff-BGM (Li et al, 2024) should also be discussed.
2. The authors mention efficiency and computational costs multiple times in the ablations but do not provide quantitive results like latency or throughput.
3. The writing of this paper is poor. The main paper is not self-contained as Section 5.6 relies on a figure in the appendix. There are also many typos.
4. For the qualitative result, it is improper to conclude that CMT has no high-frequency components. If the vertical axis is the frequency in Hz, 8K Hz is far beyond the frequency range of common instruments. For instance, the maximum frequency of a piano is usually 4K Hz. High-frequency components in the figure may be due to harmonic waves or overtones related to specific sound fonts. It is also difficult to conclude from the figure that M2UGen has repetitive structures.

**Questions:**

1. How to deal with the non-music segments? Though the vocal tracks have been removed, there might still be non-music sounds that take up to 50% (based on Line 996) of the frames.
2. It is better to evaluate the data curation process quantitively, e.g. visualize the distribution of the ImageBind-AV scores of each subset.
3. Line 477: not clear why this is related to limited GPU resources.
4. What are the total training time and number of GPUs? Given the large amount of the dataset, the training cost may be a potential concern.
5. Typos:
   1. Tables 4 and 5 are overlapped which hinders reading.
   2. Table 3: In "coverage", the bold one should be VidMuse-CAQ_LS instead of VidMuse.
   3. Line 242: the average length of music in the finetuning set is therefore 18 minutes, much longer than the other two subsets. Please double-check.
   4. Line 50: Fig.3 -> Fig. 1.
   5. Line 129: duplicate references for Gemmeke et al., 2017.
   6. Figure 2 (a): better to use larger text and change some of the text directions for better reading.
   7. Line 284: redundant ")".
   8. Line 286: ". Because" -> ', because'.
   9. Line 399: botch -> both.
   10. Line 485: visual -> visual encoder.

---

### Official Review · Reviewer_LKtR · 2024-10-30

**Soundness:** 3
**Presentation:** 3
**Contribution:** 2
**Rating:** 6
**Confidence:** 4

**Summary:**

This paper introduces VidMuse, an autoregressive model for generating music from video. The authors constructed a new large-scale dataset comprising 360k video-music pairs, which combines newly collected data with filtered existing music video data. The data construction and processing methods are described in detail. The model employs an autoregressive transformer decoder that generates music tokens directly, conditioned on long-term and short-term visual embeddings. Experimental results show that training on the new dataset and using LST fusion improves performance on both quantitative and qualitative metrics. The model is compared extensively with state-of-the-art models, and ablation studies are conducted. Demo videos are available on a webpage.

**Strengths:**

- One of the main contributions is the new constructed dataset. The dataset construction, filtering, preprocessing and usage are well-explained. The ethical statement section is critical and it addresses most of my concerns regarding a new music-video dataset.
- The long-short-term visual feature module is well-motivated. The paper is generally well-written. It clearly describes each component of VidMuse and explains the module selection with accompanied experimental results.
- Extensive demo videos are available in the anonymous webpage and supplementary material contains healthy additional details.

**Weaknesses:**

- While music source separation can be used to remove the vocal soundtrack, the final generated music will not contain any vocal music. It is not a wrong choice but human vocal music is missing and obviously it is still playing a critical role in music. It just needs more investigation to generate both background music and reasonable vocal music. Besides, an evaluation/analysis for sound separation is needed. i.e., how well does the music sound separation (demucs) work for the collected data?
- The audio in new collected video data actually contains more than music. Most movie trailers contain sound effects that are not necessary music. Unfortunately these sound effects will not be removed by music sound separation and it is unclear whether these sound effects will affect the quality.
- The technical contribution regarding model component is relatively limited. The design of long-short term visual feature fusion is straightforward but in general I am okay with that for an application + dataset paper.
- An analysis of music genre available in the dataset is missing. This is even more important than the video genre.
- It seems like a pre-trained Encodec model is used in this work but why not fine-tune or even training a new Encodec model on this new dataset?
- Although the effort and focus here is to generate music from video input only, it actually makes sense to incorporate additional text for better style control (like V2Meow). Since VidMuse leverages pre-trained MusicGen which is already a text-to-music model, why not maintaining its original capability while incorporating Long-short-term visual embeddings? This will potentially unlock more flexible applications.
- As authors mentioned already, the ImageBind score is definitely not the best choice for video-music relevance. Even training a video-music contrastive learning model on top of this new dataset would be a better choice.
- Some minor comments:
  - Font size in Figure 2 is too small.
  - Table 4 and Table 5 have bad overlapping
  - Several videos (e.g., movie trailers) in the demo webpage seem like copyright protected. My impression is it might be ok for paper reviewing stage but please follow the correct guidance and use them carefully.

**Questions:**

- Did author consider visual tokens such as VQ-GAN tokens? Or combinations of different types of visual features?
- What are GFLOPs for state-of-the-art models?

---

### Official Review · Reviewer_7cYx · 2024-10-30

**Soundness:** 3
**Presentation:** 1
**Contribution:** 2
**Rating:** 5
**Confidence:** 5

**Summary:**

VidMuse proposes a video-to-music generation framework based on a new dataset, V2M, comprising over 360,000 video-music pairs. This model utilizes a Long-Short-Term Visual Module (LSTV-Module) to integrate both global and local video features, thereby generating music that aligns semantically and emotionally with the input video. Experiments demonstrate VidMuse’s advantages over baseline models, and the authors conduct a subjective user study as part of their evaluation.

**Strengths:**

1)The V2M dataset, with extensive filtering and validation steps, is a valuable resource for future video-to-music generation studies.

2)By combining short- and long-term visual contexts, VidMuse offers an approach that could enhance alignment in generated music, potentially beneficial in diverse video genres.

**Weaknesses:**

1)I don't think this paper is well-writen and authors may write this paper in a rush, could refer to: Line 378-390 The Table 2 right side is out of the boudary of the page; Line 498-505 The Table 4 and Table 5 overlap with other in a conflict manner.

2)The evaluation is limited to the V2MBench, which is author self-proposed benchmark, and does not include any external validations on other datasets, casting doubt on its generalizability and overfit on their own dataset.

3)Metrics like FAD, KL, and ImageBind Score do not provide enough insight into real-world applications, as they lack clear explanation of their relevance to audio-visual coherence. So in the human user study should include some subjective quesions about how they rate them.

4)They didn't specify involved human objects source and how they pay them for both data quality process and user study period.

5)Overflowing tables (e.g., Tables 2, 4 and 5) affect readability and reduce the professional presentation of the work.

**Questions:**

1)How do you ensure that the music generated does not overfit specific video genres in the V2M dataset?

2)Have any experiments been conducted to compare VidMuse’s performance on other, pre-existing video-to-music datasets, except your own benchmark?

3)What is the different between the pre-train & fine-tuning stage in your method, could you specify?

**Details Of Ethics Concerns:**

This paper didn't specify who are the human subjects participate in the user study and data quality filtering and how they paid them.

---

### Official Review · Reviewer_uyZr · 2024-11-04

**Soundness:** 3
**Presentation:** 3
**Contribution:** 2
**Rating:** 3
**Confidence:** 5

**Summary:**

This paper proposes Vidmuse, a video-to-music generation framework that generates high-fidelity music in sync with visual content. The authors also propose a large-scale video-to-music dataset containing 360k video-music pairs and a new benchmark V2M-bench. The proposed framework outperforms several previous methods both on subjective and objective metrics.

**Strengths:**

1. Video-music paired datasets are scarce. The proposed high-quality, large-scale dataset benefits the community. The authors also design a reasonable and effective coarse-to-fine filtering pipeline to ensure data quality. The proposed benchmark also helps the validation of video-to-music models.

2. The proposed framework is intuitive and easy to understand. Incorporating several pretrained models (Clip, Encodec, and MusicGen transformer), the proposed method achieves state-of-the-art performance on several metrics.

3. The writing is clear and easy to follow.

**Weaknesses:**

1. I am curious about the role of the video-to-music generation task. Though some previous advances tackle the task of video-to-music generation, additional constraints are attached to these models to make them more applicable. For example, some previous works [1-6] explore the rhythm synchronization of music and video, which can generate musical soundtracks with high audio-visual rhythm correspondence [1-5], and some other previous advances generate background music with corresponding emotional responses [7] or combine the music with additional audio effects [8]. However, the proposed model seems to only be able to generate semantic-matched music, which can be easily fulfilled in a training-free way, especially considering the proposed method directly leverages the pre-trained MusicGen as the music generator. There are at least three ways to achieve a similar goal: 1). Use some video-music model (such as M2UGen [9]) to generate musical captions and then leverage MusicGen to generate semantic-matched background music. 2) Use a video captioner to generate video captions and transform them into musical captions based on its semantic information using LLM, and then leverage MusicGen to generate semantic-matched background music. 3) Use Imagebind-av, the very same model that the authors use to construct the dataset, to retrieve music with the same semantics as the visual contents, and use music captioner to generate music captions, then leverage MusicGen to generate semantic-matched background music. In other words, generating semantic-matched music, especially leveraging several existing modules, seems to be an unnecessary need, which can be solved in a training-free manner, using almost the same pretrained models. From another perspective, a good soundtrack for a given video should respond timely to the semantic change in the visual contents, yet I cannot find any explicit control module in the model architecture, nor the musical rhythm change in the provided demos. What will the music be like when the video's rhythm of the former part is rapid and enthusiastic, yet suddenly becomes slow and sad in the latter part? Consequently, the restricted applicability of the proposed model significantly diminishes the paper's contribution.

2. The model architecture is trivial. Clip is used for visual encoding, Encodec is utilized for audio codec, and MusicGen is used for music generation. That is to say, only the long-short-term visual module is the newly proposed module, while it is constructed by several attention-based integration and fusion blocks. The entire framework is more likely to be a successful industrial product rather than a highlighted research finding.

3. The experiments are insufficient. The authors only conduct experiments on some weak baseline methods. For example, VM-Net and CMT are works published 7 and 3 years ago, and M2UGen is a music-centric multi-task model that is not specifically designed for video-to-music generation. On the contrary, some newly proposed video-to-music generation methods [1-6] are not compared. Besides, have the authors tested the model's performance on other existing benchmarks, such as BGM909 [5], LORIS [3], or SymMV [6]? Experiments on more available benchmarks and comparisons with more recent advances are needed to support the authors' claim.

Reference:

[1]: Zhu, Ye, et al. "Quantized gan for complex music generation from dance videos." European Conference on Computer Vision. Cham: Springer Nature Switzerland, 2022.
[2]: Zhu, Ye, et al. "Discrete contrastive diffusion for cross-modal music and image generation." arXiv preprint arXiv:2206.07771 (2022).
[3]: Yu, Jiashuo, et al. "Long-term rhythmic video soundtracker." International Conference on Machine Learning. PMLR, 2023.
[4]: Su, Kun, et al. "V2Meow: Meowing to the Visual Beat via Video-to-Music Generation." Proceedings of the AAAI Conference on Artificial Intelligence. Vol. 38. No. 5. 2024.
[5]: Li, Sizhe, et al. "Diff-BGM: A Diffusion Model for Video Background Music Generation." Proceedings of the IEEE/CVF Conference on Computer Vision and Pattern Recognition. 2024.
[6] Zhuo, Le, et al. "Video background music generation: Dataset, method and evaluation." Proceedings of the IEEE/CVF International Conference on Computer Vision. 2023.
[7] Kang, Jaeyong, Soujanya Poria, and Dorien Herremans. "Video2music: Suitable music generation from videos using an affective multimodal transformer model." Expert Systems with Applications 249 (2024): 123640.
[8] Movie Gen: A Cast of Media Foundation Models, meta, 2024
[9] Liu, Shansong, et al. "M $^{2} $ UGen: Multi-modal Music Understanding and Generation with the Power of Large Language Models." arXiv preprint arXiv:2311.11255 (2023).

**Questions:**

My major concerns are listed in the weaknesses part mentioned above, and I only have minor questions here.

1. For the dataset composition, there are 400K videos derived from YouTube and IMDB, what is the proportion? What kind of query set is adopted to retrieve the videos?

2. Why does the model perform worse when using MusicGen-large as the decoder? In the manuscript, it says 'this discrepancy can be partly attributed to limited GPU resources', can the model be trained using some parameter-efficient training strategy such as LoRA?

3. For the model architecture, why the music token decoder is involved in training considering that the vanilla MusicGen is able to generate high-fidelity music? Maybe adopting a trainable linear projection layer to the decoder could significantly reduce the model parameter and solve the training difficulty of MusicGen-large.

4. Table 4 is overlapped with Table 5, please consider adjusting the table spacing.

---

### Note · Authors · 2024-11-14

I have read and agree with the venue's withdrawal policy on behalf of myself and my co-authors.